# Two-Stage Robust Optimal Scheduling of Flexible Distribution Networks Based on Pairwise Convex Hull

**Haiyue Yang [1,2], Shenghui Yuan [1], Zhaoqian Wang [3] and Dong Liang [3,\*]**

[1] State Grid Hebei Electric Power Company Hengshui Power Supply Company, Hengshui 148530, China
[2] Hengshui Electric Power Design Co., Ltd., Hengshui 148530, China
[3] State Key Laboratory of Reliability and Intelligence of Electrical Equipment, Hebei University of Technology, Tianjin 300401, China; duliuxuejia@163.com
\* Correspondence: liangdong@hebut.edu.cn

**Abstract:** With distributed generation (DG) being continuously connected into distribution networks, the stochastic and fluctuating nature of its power generation brings ever more problems than before, such as increasing operating costs and frequent voltage violations. However, existing robust scheduling methods of flexible resources tend to make rather conservative decisions, resulting in high operation costs. In view of this, a two-stage robust optimal scheduling method for flexible distribution networks is proposed in this paper, based on the pairwise convex hull (PWCH) uncertainty set. A two-stage robust scheduling model is first formulated considering coordination among on-load tap changers, energy storage systems and flexible distribution switches. In the first stage, the temporal correlated OLTCs and energy storage systems are globally scheduled using day-ahead forecasted DG outputs. In the second stage, FDSs are scheduled in real time in each time period based on the first-stage decisions and accurate short-term forecasted DG outputs. The spatial correlation and uncertainties of the outputs of multiple DGs are modeled based on the PWCH, such that the decision conservativeness can be reduced by cutting regions in the box with low probability of occurrence. The improved column-and-constraint generation algorithm is then used to solve the robust optimization model. Through alternating iterations of auxiliary variables and dual variables, the nonconvex bilinear terms induced by the PWCH are eliminated, and the subproblem is significantly accelerated. Test results on the 33-bus distribution system and a realistic 104-bus distribution system validate that the proposed PWCH-based method can obtain much less conservative scheduling schemes than using the box uncertainty set.

**Keywords:** flexible distribution network; flexible distribution switch; pairwise convex hull; column-and-constraint generation; robust optimization

## 1. Introduction

With distributed generation (DG) being continuously connected into distribution networks, the stochastic and fluctuating nature of its power generation brings ever more problems than before, such as increasing operating costs, voltage violations, etc. Therefore, exploiting the potential of flexible resources to improve the DG penetration level and reduce the carbon emission level is urgent.

Dispatchable flexible resources, such as mechanical switches, on-load tap changers (OLTCs) and capacitor banks in traditional distribution networks, can neither be fast nor continuously adjusted. At the same time, they cannot be frequently controlled due to the life span reason, which limits their ability to cope with the increasing integration of DG. In recent years, the back-to-back voltage source converter (VSC)-based flexible distribution switch (FDS) has picked up momentum to overcome the drawbacks of mechanical devices [1,2]. The replacement of traditional mechanical switches with FDSs converts traditional distribution networks into flexible distribution networks (FDN), where fast and accurate power flow control can be realized.

The scheduling problem of FDNs has been studied for years. In [3], a coordinated voltage and VAR control method based on FDSs is proposed for distribution networks to minimize operation costs and eliminate voltage violations. The nonconvex mixed-integer nonlinear optimization model is converted into a mixed-integer second-order cone programming (MISOCP) model, which can be efficiently solved. In [4], an FDS-based operation strategy for unbalanced distribution networks is proposed to simultaneously reduce power losses and mitigate the three-phase unbalance problem. The nonconvex, nonlinear optimization model is converted into a semidefinite programming formulation.

However, deterministic scheduling neglects the uncertainty of load demands and intermittent DG outputs and may lead to inappropriate control schemes and even result in the violation of security constraints. Thus, it is required that these uncertainties are dealt with to ensure the feasibility and reliability of the system operation. Current research on dispatch methods of distribution networks with DG uncertainties mainly fall into two categories, i.e., stochastic optimization (SO) and robust optimization (RO) [5–7]. Compared with SO, which depends on the probability density function of uncertain parameters, the only information needed for RO is the ranges of uncertain parameters, which are much easier to obtain in reality. In [8], a two-stage RO model is proposed to coordinate the OLTC ratios, reactive power compensators and charge–discharge power of energy storage systems (ESSs). The column-and-constraint generation (CCG) algorithm is applied to solve the two-stage RO model. In [9], a two-stage robust optimal dispatch model is presented for an islanded AC/DC hybrid microgrid, where the first stage determines the startup/shutdown state of the diesel engine generator and the operating state of the bidirectional converter of the microgrid, while the second stage optimizes the power dispatch of individual units in the microgrid. In [10], an RO model is established for AC/DC distribution networks to co-optimize the slopes of active and reactive power droop control of VSCs, with the aim to minimize the total network loss while ensuring the system security. In [11], a two-stage RO model is built for day-ahead dispatching of FDSs to tackle the uncertainties of PV outputs, eliminate voltage violations and reduce power losses. However, the FDS is considered to be the only flexible resource, while the coordination with other control devices is neglected. In [12], a bilevel RO model is proposed for the service restoration problem of FDNs to obtain the optimal service restoration scheme, i.e., the switch statuses and range of power transmitted by FDS terminals.

Uncertain parameters in RO are usually modeled using uncertainty sets such as box, ellipsoid, polyhedron and convex hull. The box uncertainty set cannot characterize the correlation among random variables, and the results tend to be conservative. The polyhedral uncertainty set (or the budget uncertainty set) and the ellipsoidal uncertainty set can take the correlation among uncertain parameters into account and at the same time have a linear structure; thus, they are widely used [5–7,9,11]. However, polyhedral and ellipsoidal uncertainty sets cannot characterize the nonlinear correlation of uncertain parameters. Furthermore, the ellipsoidal uncertainty set converts linear constraints into quadratic constraints and the original linear programming model is transformed into a quadratic programming model and the complexity increases a lot. In [13], an RO-based economic dispatch method is proposed for active distribution networks based on extreme scenarios to adapt to historical data sets and reduce the decision conservativeness. However, it still assumes that the uncertainty data lie in a specified uncertainty set and the complex, asymmetric correlation among uncertain parameters cannot be handled. The convex hull is the smallest convex set that can cover the historical data set and enjoys the least decision conservativeness. However, it has limited application due to excessive linear constraints, large computational effort and computational complexity in case of high-dimensional data. To this end, the pairwise convex hull (PWCH) is proposed in [14,15], which is computationally efficient and is linearly expensive for high-dimensional data.

A comparative table on existing RO-based scheduling methods is shown in Table 1, in the aspects of first-stage decisions, second-stage decisions, uncertainty set and solution algorithm, to give a clearer clarity to this state-of-the-art method.

**Table 1.** Comparison of existing RO-based scheduling methods.

| Reference | First Stage Decision | Second Stage Decision | Uncertainty Set | Solution Algorithm |
|---|---|---|---|---|
| [5] | Network topology | \ | Polyhedron | CCG |
| [6] | Network topology | \ | Polyhedron | CCG |
| [7] | Network topology, reactive output of VAR compensators and OLTC ratios | DG installation capacity | Polyhedron | CCG |
| [8] | OLTC ratios, discrete VAR compensators and charge–discharge power of ESSs | Continuous VAR compensators | Box | CCG |
| [9] | Startup/shutdown state of diesel engine generator, operating state of the converters | Individual units | Polyhedron | CCG |
| [10] | Slopes of power droop control of VSCs | \ | Box | CCG |
| [11] | Power injection of soft open points | \ | Polyhedron | CCG |
| [12] | Network topology and power injection of soft open points | Power injection of soft open points | Box | CCG |
| [13] | Switching capacitor and OLTC ratios | SVG | Data-adaptive polyhedron | Extreme Scenario |
| [15] | Non-AGC units | AGC units | PWCH | CG |
| [16] | Capacitor banks and OLTC ratios | PV inverters | Polyhedron | CCG |

In this paper, a two-stage robust optimal scheduling method is proposed for FDNs. The contributions are summarized as follows:

(1) A two-stage robust optimal scheduling model is formulated for FDNs, considering coordination among OLTCs, ESSs and FDSs. In the first stage, the temporal correlated OLTCs and ESSs are globally scheduled using day-ahead forecasted DG outputs, while in the second stage, FDSs are scheduled in real time in each time period, based on the first-stage decisions and accurate short-term forecasted DG outputs.

(2) The spatial correlation of the outputs of multiple DGs is modeled based on the PWCH, such that the high-dimensional convex hull is relaxed into an intersection of finite PWCHs. By cutting regions in the box with low probability of occurrence, the decision conservativeness can be reduced.

(3) The improved CCG algorithm is then used to solve the RO model. Through alternating iterations of auxiliary variables and dual variables, the nonconvex bilinear terms induced by the PWCH are eliminated, and the subproblem is significantly accelerated.

The rest of this paper is organized as follows. Section 2 presents the deterministic scheduling model for FDNs. Section 3 presents the PWCH uncertainty set. Section 4 presents the proposed two-stage robust optimal scheduling model and solution algorithm for FDNs. Section 4 describes the simulation results, and conclusions are drawn in Section 5.

## 2. Deterministic Optimal Scheduling for FDNs

### 2.1. Objective Function

The objective is to minimize the daily loss of the distribution network as follows:

$$\min \sum_{t \in \Omega_T} \left( \sum_{i \in \Omega_b} P_{i,t} + \sum_{v \in \Omega_{FDS}} \sum_{i \in \Omega_{b(v)}} P_{i,t,FDS}^{loss} + \sum_{i \in \Omega_{ESS}} P_{i,t,ESS}^{loss} \right) \Delta t \tag{1}$$

where $P_{i,t}$ is the real power injection of bus $i$ during time period $t$; $P_{i,t,FDS}^{loss}$ is the VSC power loss of the multiterminal FDS near the terminal of bus $i$ during time period $t$; $P_{i,t,ESS}^{loss}$ is the charging and discharging power loss of the ESS installed at bus $i$ during time period $t$; $\Omega_T$ is the set of all time periods; $\Omega_b$ is the set of all buses; $\Omega_{FDS}$ is the set of all FDSs; $\Omega_{b(v)}$ is the set of all buses associated with the $v$th FDS; and $\Omega_{ESS}$ is the set of all buses with ESS installed; $\Delta t$ is the interval (h) of each time period.

*2.2. Constraints*

2.2.1. Steady-State Operational Constraints of FDSs

The multiterminal back-to-back FDS consists of multiple VSCs, and the control variables are the real and reactive power transmitted by each VSC, which is operated at the four-quadrant operation mode. Assuming the FDS power is positive, if it is injected into the network, then the following steady-state operational constraints need to be satisfied for the $v$th FDS:

$$P_{i,t,\text{FDS}}^{\text{loss}} = A_{\text{loss,FDS}} \sqrt{(P_{i,t,\text{FDS}})^2 + (Q_{i,t,\text{FDS}})^2}, \forall i \in \Omega_{\text{b}(v)} \tag{2}$$

$$\sum_{i \in \Omega_{\text{b}(v)}} (P_{i,t,\text{FDS}} + P_{i,t,\text{FDS}}^{\text{loss}}) = 0 \tag{3}$$

$$Q_{\text{min,FDS}} \leq Q_{i,t,\text{FDS}} \leq Q_{\text{max,FDS}}, \forall i \in \Omega_{\text{b}(v)} \tag{4}$$

$$\sqrt{(P_{i,t,\text{FDS}})^2 + (Q_{i,t,\text{FDS}})^2} \leq S_{\text{max,FDS}}, \forall i \in \Omega_{\text{b}(v)} \tag{5}$$

where $P_{i,t,\text{FDS}}$ and $Q_{i,t,\text{FDS}}$ are the real and reactive power injections of the FDS into bus $i$ during time period $t$, respectively; $A_{\text{loss,FDS}}$ is the FDS loss factor; $Q_{\text{min,FDS}}$ and $Q_{\text{max,FDS}}$ are the upper and lower limits of the reactive power through each FDS, respectively; and $S_{\text{max,FDS}}$ is the maximum apparent power allowed through the FDS. Equation (3) makes the sum of the real power injection into all associated feeders by the FDS and the power loss of the FDS come out to zero. Equation (4) makes the reactive power injection of the FDS not exceed its adjustable reactive power limit. Equation (5) makes the apparent power of the FDS not exceed its capacity.

2.2.2. Steady-State Operational Constraints of ESSs

During steady state, the following steady-state operational constraints need to be satisfied for the ESS installed at bus $i$:

$$0 \leq P_{i,t,\text{ESS}}^+ \leq P_{i,\text{max}}^+, 0 \leq P_{i,t,\text{ESS}}^- \leq P_{i,\text{max}}^- \tag{6}$$

$$E_{i,t} = E_{i,t-1} + \eta_i^c P_{i,t,\text{ESS}}^+ \Delta t - (1/\eta_i^d) P_{i,t,\text{ESS}}^- \Delta t \tag{7}$$

$$E_{i,0} = E_{i,\text{T}} \tag{8}$$

$$E_{i,\text{ESS}} \cdot SOC_{i,\text{min}} \leq E_{i,t} \leq E_{i,\text{ESS}} \cdot SOC_{i,\text{max}} \tag{9}$$

where $P_{i,t,\text{ESS}}^+$ and $P_{i,t,\text{ESS}}^-$ are the charging and discharging real power of the ESS installed at bus $i$ during time period $t$, respectively; $P_{i,\text{max}}^+$ and $P_{i,\text{max}}^-$ are the maximum charging and discharging real power of the ESS installed at bus $i$, respectively; $E_{i,t}$ is the remaining energy of the ESS installed at bus $i$ during time period $t$; $\eta_i^c$ and $\eta_i^d$ are the charging and discharging efficiency of the ESS installed at bus $i$, respectively; $E_{i,\text{ESS}}$ is the energy capacity of the ESS installed at bus $i$; and $SOC_{i,\text{max}}$ and $SOC_{i,\text{min}}$ are the maximum and minimum state of charge (SOC) of the ESS installed at bus $i$. Equation (6) makes the charging and discharging power not exceed the maximum value at any time period; Equation (7) makes the remaining energy satisfy the continuity constraint. Equation (8) makes the remaining energy at the end of each day equal to the initial energy of that day. Equation (9) makes the ESS free from deep charging or discharging.

2.2.3. Operational Constraints of OLTCs

As shown in Figure 1, an OLTC is split into series of an impedance branch $i$-$m$ and an ideal transformer branch $m$-$j$, and the following constraints need to be satisfied during operation:

$$V_{m,t} = \tau_{ij,t} V_{j,t} \tag{10}$$

$$\tau_{ij,t} = \tau_{ij,\text{min}} + N_{ij,t} \Delta \tau_{ij} \tag{11}$$

$$0 \leq N_{ij,t} \leq N_{ij,\text{max}} \tag{12}$$

$$\sum_{t \in \Omega_{\mathrm{T}}, t>1} \left| N_{ij,t} - N_{ij,t-1} \right| \le \beta_{ij,\max} \tag{13}$$

where $\tau_{ij}$ is the ratio of the OLTC branch *i-j*; $\tau_{ij,\min}$ is the minimum ratio of the OLTC branch *i-j*; $N_{ij,t}$ is the tap position of the OLTC branch *i-j* during time period *t*; $N_{ij,\max}$ is the maximum tap position number of the OLTC branch *i-j*; $\beta_{ij,\max}$ is the maximum allowed number of daily actions of the OLTC branch *i-j*; and $\Delta\tau_{ij}$ is the difference in the ratios between adjacent tap positions of the OLTC branch *i-j*. For example, if there are five upper tap positions and five lower tap positions, then $N_{ij,\max} = 11$, $N_{ij,t}$ takes the integers 0~10 and $\Delta\tau_{ij}$ takes the values 0.01~0.10.

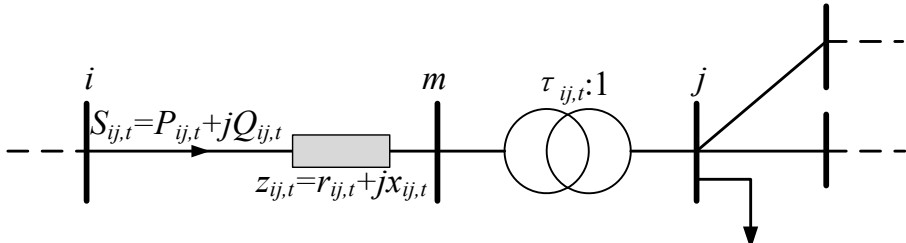

**Figure 1.** Illustration of the OLTC.

### 2.2.4. Power Flow Constraints

Using the DistFlow model, the nodal power flow balance equation should be satisfied for each non-slack bus as follows:

$$\begin{cases} P_{i,t} = P_{i,t,\mathrm{DG}} - P_{i,t,\mathrm{L}} + P_{i,t,\mathrm{FDS}} - P_{i,t,\mathrm{ESS}}^{+} + P_{i,t,\mathrm{ESS}}^{-} \\ Q_{i,t} = -Q_{i,t,\mathrm{L}} + Q_{i,t,\mathrm{FDS}} \\ \sum_{(k,i) \in \Omega_{\mathrm{l}}} \left( P_{ki,t} - I_{ij,t}^2 r_{ki} \right) - \sum_{(i,j) \in \Omega_{\mathrm{l}}} P_{ij,t} = -P_{i,t} \\ \sum_{(k,i) \in \Omega_{\mathrm{l}}} \left( Q_{ki,t} - I_{ij,t}^2 x_{ki} \right) - \sum_{(i,j) \in \Omega_{\mathrm{l}}} Q_{ij,t} = -Q_{i,t} \\ V_{j,t}^2 = V_{i,t}^2 - 2(P_{ij,t} r_{ij} + Q_{ij,t} x_{ij}) + (r_{ij}^2 + x_{ij}^2) I_{ij,t}^2 \\ P_{ij,t}^2 + Q_{ij,t}^2 = V_{i,t}^2 I_{ij,t}^2 \end{cases} \tag{14}$$

where $P_{ij,t}$, $Q_{ij,t}$ and $I_{ij,t}$ are the real and reactive power at the "from" end and current amplitude of branch *i-j* during time period *t*, respectively; $P_{i,t,\mathrm{DG}}$ and $P_{i,t,\mathrm{L}}$ are the DG real power injection and real power load at bus *i* during time period *t*, respectively; $Q_{i,t,\mathrm{DG}}$ and $Q_{i,t,\mathrm{L}}$ are the DG reactive power injection and reactive power load at bus *i* during time period *t*, respectively; $r_{ij}$ and $x_{ij}$ are the resistance and reactance of branch *i-j*, respectively; and $\Omega_{\mathrm{l}}$ is the set of all lines.

### 2.2.5. Thermal Limit Constraint

The current amplitude of each branch should not exceed the ampacity:

$$0 \le \left( I_{ij,t} \right)^2 \le \left( I_{ij,\max} \right)^2 \tag{15}$$

where $I_{ij,\max}$ is the ampacity of branch *i-j*.

### 2.2.6. Bus Voltage Constraints

The node voltage magnitude should not exceed the upper and lower bound:

$$\underline{V}_i \le V_i \le \overline{V}_i \tag{16}$$

where $\overline{V}_i$ and $\underline{V}_i$ are the upper and lower limits of the voltage magnitude of bus *i*, respectively.

### 2.3. Model Reformulation as a MISOCP Problem

Since the power flow Equation (14) is nonconvex and, at the same time, nonconvex bilinear terms exist in the OLTC constraint (10), the optimization model (1)~(16) is a nonconvex mixed-integer nonlinear programming problem, which is difficult to solve. Therefore, this section converts the original problem into a MISOCP model which can be efficiently solved.

#### 2.3.1. Reformulation of the Power Flow Constraints

First, auxiliary variables $w_{ij,t}$ and $u_{i,t}$ are introduced to replace $I_{ij,t}^2$ and $V_{i,t}^2$, and with the help of the big-M relaxation technique, the last two equations in Equation (13) become:

$$\begin{cases} \left\| [2P_{t,ij}\, 2Q_{t,ij}\, l_{t,ij} - u_{t,ij}]^{\mathrm{T}} \right\|_2 \leq w_{t,ij} + u_{t,ij} \\ u_{i,t} - u_{j,t} = 2(P_{ij,t}r_{ij} + Q_{ij,t}x_{ij}) - (r_{ij}^2 + x_{ij}^2)w_{ij,t} \\ 0 \leq w_{ij,t} \leq (I_{ij,\max})^2 \end{cases} \tag{17}$$

where $M$ is a larger positive number.

#### 2.3.2. Reformulation of the FDS Constraints

The FDS capacity constraint (5) can be relaxed as the following cone constraint:

$$\left\| [P_{i,t,\mathrm{FDS}} Q_{i,t,\mathrm{FDS}}]^{\mathrm{T}} \right\|_2 \leq P_{i,t,\mathrm{FDS}}^{\mathrm{loss}} / A_{\mathrm{loss,FDS}} \tag{18}$$

#### 2.3.3. Reformulation of the OLTC Constraints

Using the square of the voltage magnitude and the square of the OLTC ratio, constraint (10) becomes:

$$u_{m,t} = \tau_{ij,t}^2 u_{j,t} \tag{19}$$

For (11), the binary variable $B_{ij,s,t}$ is introduced as the flag of the $s$th tap position for the OLTC branch $i$-$j$ during time period $t$. Then, $N_{ij,t}$ can be expressed as the cumulative sum of the $B_{ij,s,t}$ for each tap position, and the OLTC constraints become:

$$N_{ij,t} = \sum_{s=1}^{N_{ij,\max}} B_{ij,s,t}, \; B_{ij,s,t} \in \{0,1\} \tag{20}$$

$$B_{ij,s,t} - B_{ij,s-1,t} \leq 0 \tag{21}$$

$$u_{m,t} = \left( \tau_{ij,\min}^2 + \sum_{s=1}^{N_{ij,\max}} B_{ij,s,t}\Delta\tau_{ij,s}^2 \right) u_{j,t} = \tau_{ij,\min}^2 u_{j,t} + \sum_{s=1}^{N_{ij,\max}} B_{ij,s,t}u_{j,t}\Delta\tau_{ij,s}^2 \tag{22}$$

where $B_{ij,s,t}$ is the flag of the $s$th tap position for the OLTC branch $i$-$j$ during time period $t$; $B_{ij,s,t} = 1$ indicates that the tap position $s$ is lower or equal to the actual position; and $B_{ij,s,t} = 0$ indicates that the tap position $s$ is higher than the actual position. The values of $\Delta\tau_s$ and $\Delta\tau_s^2$ for different tap position $s$ are shown in Table 2.

**Table 2.** OLTC tap positions.

| s | 0 | 1 | 2 | 3 | 4 | 5 | 6 | 7 | 8 | 9 | 10 |
|---|---|---|---|---|---|---|---|---|---|---|----|
| $\tau_s$ | 0.95 | 0.96 | 0.97 | 0.98 | 0.99 | 1 | 1.01 | 1.02 | 1.03 | 1.04 | 1.05 |
| $\Delta\tau_s$ | / | 0.01 | 0.01 | 0.01 | 0.01 | 0.01 | 0.01 | 0.01 | 0.01 | 0.01 | 0.01 |
| $\tau_s^2$ | 0.9025 | 0.9216 | 0.9409 | 0.9604 | 0.9801 | 1 | 1.201 | 1.404 | 1.609 | 1.0816 | 1.1025 |
| $\Delta\tau_s^2$ | / | 0.0191 | 0.0193 | 0.0195 | 0.0197 | 0.0199 | 0.0201 | 0.0203 | 0.0205 | 0.0207 | 0.0209 |

Constraint (22) still contains a nonconvex bilinear term $B_{ij,s,t}u_{j,t}$, so we introduce $\lambda_{ij,s,t} = B_{ij,s,t}u_{j,t}$. Using the big-M relaxation technique, (22) becomes:

$$u_{m,t} = \tau_{ij,\min}^2 u_{j,t} + \sum_{s=1}^{N_{ij,\max}} \lambda_{ij,s,t} \Delta \tau_{ij,s}^2 \tag{23}$$

$$0 \leq \lambda_{ij,s,t} \leq M \cdot B_{ij,s,t} \tag{24}$$

$$0 \leq u_{j,t} - \lambda_{ij,s,t} \leq M \cdot (1 - B_{ij,s,t}) \tag{25}$$

The final OLTC constraints include (12)–(13), (20)–(21) and (23)–(25), all of which are easily handled linear constraints.

At this point, the deterministic optimal scheduling model is transformed into a MIS-OCP problem, which can be efficiently solved using a commercial solver.

## 3. The PWCH Uncertainty Set

### 3.1. The Convex Hull Uncertainty Set

Assume that $D$ buses in the distribution network are installed with DG, and $N$ historical data points (scenarios) are recorded in history for each DG bus. The $i$th scenario can be represented as a $D$-dimensional column vector $\boldsymbol{u}_i = [u_{i,1}, u_{i,2}, \dots, u_{i,D}]^{\mathrm{T}} \in \mathbb{R}^D$, i.e., a point in the $D$-dimensional Euclidean space. Then, the $N$ historical scenarios can be represented as a high-dimensional point set $\Omega_u = \{u_1, u_2, \dots, u_N\}$, consisting of $N$ points in the $D$-dimensional Euclidean space. A high-dimensional convex hull enclosing all the points can be constructed as:

$$\mathcal{H} = \left\{ \boldsymbol{u} \in \mathbb{R}^D \middle| A\boldsymbol{u} \leq \boldsymbol{b}; A \in \mathbb{R}^{M \times D}, \boldsymbol{b} \in \mathbb{R}^M \right\} \tag{26}$$

The number of hyperplanes $M$ of this convex hull will grow exponentially with increasing $D$. If it is used for RO, the model solving will suffer a huge computational burden.

### 3.2. The PWCH Uncertainty Set

The idea of the PWCH uncertainty set is that the points in the $D$-dimensional Euclidean space are projected to different two-dimensional (2D) planes, which will yield $C_2^D = D(D-1)/2$ axial PWCHs. Then, the intersection of all PWCHs can be used as the outer approximation of the original convex hull, as shown in Figure 2 ($D = 3$), where the intersection of the red, black and blue PWCHs can be used as the outer approximation of the original convex hull. Specifically, after projecting to the 2D plane corresponding to the $m$th and the $n$th dimension ($1 \leq m < n \leq D$), a 2D convex hull can be constructed for the set of points in the 2D plane, and the range of all dimensions except $m$ and $n$ is relaxed to $(-\infty, +\infty)$, denoted as

$$\mathcal{H}^{(m,n)} = \left\{ \boldsymbol{u} \in \mathbb{R}^D \middle| A^{(m,n)} \boldsymbol{u}^{(m,n)} \leq \boldsymbol{b}^{(m,n)}, -\infty < [\boldsymbol{u}]_k < +\infty \text{ for } k \notin \{m, n\} \right\} \tag{27}$$

where $\boldsymbol{u}$ and $\boldsymbol{u}^{(m,n)}$ are the scenario vectors in the $D$-dimensional Euclidean space and the 2-D vectors after projection in the 2D plane corresponding to the $m$th and the $n$th dimension, respectively, and $A^{(m,n)}$ and $\boldsymbol{b}^{(m,n)}$ are the coefficient matrices and the right-side vector of the to the linear inequality constraint for the corresponding 2D convex hull, respectively.

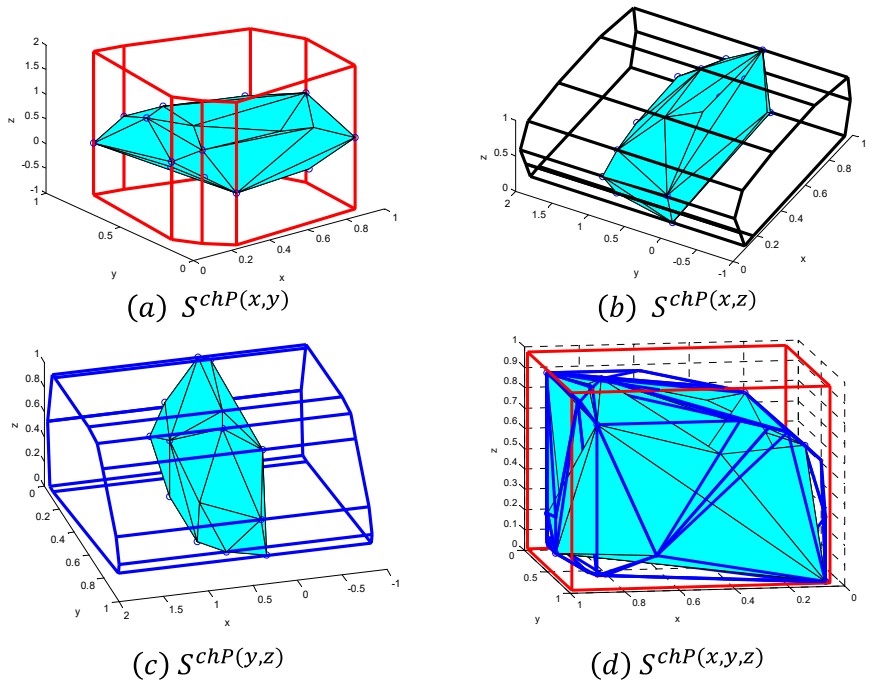

**Figure 2.** Illustration of the PWCH uncertainty set.

It can be proved that the original high-dimensional convex hull is contained within the intersection of all PWCHs, and this outer-approximation is contained within a *D*-dimensional box, i.e.,

$$\mathcal{H} \subseteq \left\{ \mathcal{H}^{\text{PWCH}} = \underset{(m,n) \in \{(m,n) | \forall 1 \leq m < n \leq D\}}{\cap} \mathcal{H}^{(m,n)} \right\} \subseteq \mathcal{H}^{\text{Box}} \tag{28}$$

The PWCH can relax the original high-dimensional convex hull into the intersection of finite PWCHs, which can significantly reduce the number of constraints and the computational effort when used for RO.

## 4. Two-Stage Robust Optimal Scheduling for FDNs

### 4.1. The Two-Stage Framework

Based on the deterministic optimal dispatching model, a two-stage robust optimal scheduling model is established in this section using the PWCH uncertainty set. In the first stage, OLTC action strategy and ESS charging/discharging power are used as control variables to solve a multiperiod coupled model for the whole day, and the first-stage decisions are passed to the next stage as a fixed value. In the second stage, the real and reactive powers through the FDS are used as control variables for fast and continuous power regulation based on more accurate ultra-short-term forecasted DG outputs. The control variables in the first stage are selected due to the following considerations:

(1) the ESS states and the OLTC tap positions are temporally coupled in different time periods;

(2) Some ESSs do not belong to the utility and their charging/discharging power should be determined in advance;

(3) OLTC is a slow-acting device that should avoid being controlled in real time.

Therefore, global optimization is required for the first stage by integrating the day-ahead DG prediction information of all time periods throughout the day and making decisions of ESS charging/discharging power and OLTC action strategies before the next day comes. In addition, during the first stage, the real-time response capability of FDSs to DG output uncertainty in the intraday redispatch stage should also be considered, and

FDS real and reactive power are also involved as control variables in the first stage, but the solved FDS power need not be passed to the second stage.

*4.2. Mathematical Formulation*

For simplicity of clarity, the RO model is built as the following compact form:

$$
\min_{x} \max_{d \in \mathcal{H}^{\mathrm{PWCH}}} L(x, d) = \min_{x} \max_{d \in \mathcal{H}^{\mathrm{PWCH}}} \min_{y} \sum_{i \in \Omega_{\mathrm{b}}} f
$$
$$
s.t. \begin{cases} x \in \mathcal{X} \\ y \in \mathcal{Y}(x, d), \ \forall d \in \mathcal{H}^{\mathrm{PW-CH}} \end{cases}
\tag{29}
$$

where $x$ is the first-stage control variable vector; $y$ is the joint vector of the second-stage control variables and the second-stage state variables; $d$ is the scenario variable vector; $L(x,d)$ is the objective function under the first-stage decision $x$ and the scenario $d$; $\mathcal{X}$ is the set of all feasible day-ahead decisions, including OLTC action strategies and ESS charging/discharging power during each period; and $\mathcal{Y}(x, d)$ is the set of all feasible second-stage solutions under the first-stage decision $x$ and the scenario $d$, defined as

$$
\mathcal{Y}(x, d) := \left\{ y \ \middle| \ \begin{array}{l} Dy \geq f - Ax \\ Cy = d \\ \|Gy\|_2 \leq g^{\mathrm{T}} y \end{array} \right\}
\tag{30}
$$

where the three equations present the linear inequality constraints, the linear equality constraints and the second-order cone constraints, respectively; $D, f$ and $A$ are the coefficient matrices and right-side vector after all linear inequality constraints are rewritten into the matrix-vector form; $C$ is the coefficient matrix after all linear equality constraints are rewritten into the matrix-vector form; and $G$ and $g$ are the coefficient matrix and vector after all second-order cone constraints are rewritten into the matrix-vector form.

*4.3. Solution Algorithm*

Using the CCG algorithm framework [17], the model can be divided into a master problem and a subproblem, both of which are MISOCP problems. The master problem solves the first-stage decisions, considering the constraints of the worst-case scenarios returned by the subproblem, and updates the lower bound of the objective function. The subproblem solves the worst-case scenario under the first-stage decisions, returns the worst-case scenario back to the master problem, and updates the upper bound of the objective function. The master problem is:

$$
\min_{x, y^{(s)}, Lt \in \Omega_{\mathrm{T}}} \sum L
$$
$$
s.t. \begin{cases} x \in \mathcal{X} \\ L \geq L(x, d^{(s)}), \ \forall s = 1, \cdots, k \\ y^{(s)} \in \mathcal{Y}(x, d^{(s)}), \ \forall s = 1, \cdots, k \\ L \geq 0 \end{cases}
\tag{31}
$$

where $k$ is the number of the worst-case scenarios returned by the subproblem, and is also used to indicate the current iteration index.

Each time the master problem is solved, the lower bound of the objective function is updated, and the first-stage OLTC and ESS decisions are passed to the subproblem. In the second stage, the control variable is only the FDS power, so the subproblem is no longer temporally coupled among time periods and can be solved for each time period in parallel.

The objective function of the subproblem is the sum of objectives for all time periods. The form of the subproblem is shown as follows:

$$L(x) := \sum_{t \in \Omega_T} \max_{d \in \mathcal{H}^{PWCH}} \min_{y} b^T y$$

$$s.t. \begin{cases} Dy \geq f - Ax^* & (\lambda) \\ Cy = d & (\pi) \\ \|Gy\|_2 \leq g^T y & (\sigma, \mu) \end{cases} \tag{32}$$

where $b$ is the coefficient vector after the objective function is rewritten in the matrix-vector form; $\pi, \lambda, \sigma$ and $\mu$ are the Lagrange multiplier (dual variables) vectors of the corresponding constraints; and the superscript * indicates the results of the master problem.

Since the subproblem is a max–min problem, the min problem needs to be transformed into the following max problem by constructing a Lagrangian function. The transformed model is as follows:

$$L(x^*) := \max_{d, \pi, \lambda, \sigma, \mu} (f - Ax^*)^T \lambda + d^T \pi$$

$$s.t. \begin{cases} D^T \lambda + C^T \pi + \sum (G^T y + g^T \mu) = b \\ \|\sigma\|_2 \leq \mu \\ \pi, \mu \geq 0 \end{cases} \tag{33}$$

Once the subproblem is solved, the lower bound of the objective function is updated. If the termination condition is not satisfied, the solved worst-case scenario is returned to the master problem, and a group of variables and constraints for this worst-case scenario are added to the master problem.

*4.4. Subproblem Solution Algorithm*

It is noted that there is a nonconvex bilinear term $d^T \pi$ in the objective function of model (33). For the nonconvex bilinear term, although the Gurobi 9 solver succeeds in solving the model solution through the spatial branching method, it is still rather time-consuming. Therefore, we decompose the subproblem into a linear programming (LP) problem and a second-order cone programming (SOCP) problem by alternating direction iteration of auxiliary and dual variables.

The main parts of the algorithm are listed as follows, where, during the second stage, the outer loop is solved sequentially for 24 time periods, and the inner loop is an alternating direction iteration for each time period:

(1) Use the forecasted scenario as the initial scenario in the master problem, i.e., set $d^* = d_0$. Set the upper bound of the objective function as a larger number.

(2) Fix the dual variables ($\pi$, $\lambda$, $\delta$ and $\mu$), solve the following inner LP problem, update the upper bound of the inner loop, and pass the solved auxiliary variable $d$ to the inner SOCP problem:

$$L(x^*, d_0) := \max_{\pi^*, \lambda^*, \sigma^*, \mu^*} (f - Ax^*)^T \lambda + d_0^T \pi^*$$

$$s.t. \quad D^T \lambda^* + C^T \pi^* + \sum (G^T y + g^T \mu^*) = b \tag{34}$$

(3) Fix the auxiliary variable $d$, solve the following inner SOCP problem, update the lower bound of the inner loop and pass the solved dual variables ($\pi$, $\lambda$, $\delta$ and $\mu$) to the inner LP problem:

$$L(x^*, d^*) := \max_{\pi, \lambda, \sigma, \mu} (f - Ax^*)^T \lambda + d^{*T} \pi$$

$$s.t. \begin{cases} D^T \lambda + C^T \pi + \sum (G^T y + g^T \mu) = b \\ \|\sigma\|_2 \leq \mu \\ \pi, \mu \geq 0 \end{cases} \tag{35}$$

(4) If the inner loop iteration for time period $t$ converges, solve the next time period until all time periods are completed; otherwise, return to step (2).

If all 24 time periods are completed (i.e., the subproblem is completed), the upper bound of the outer loop is updated as the sum of the 24 problems' optimal objective values, and the worst-case scenario solved by the subproblem is passed to the master problem for the next outer iteration. The overall flowchart of the improved CCG algorithm is shown in Figure 3.

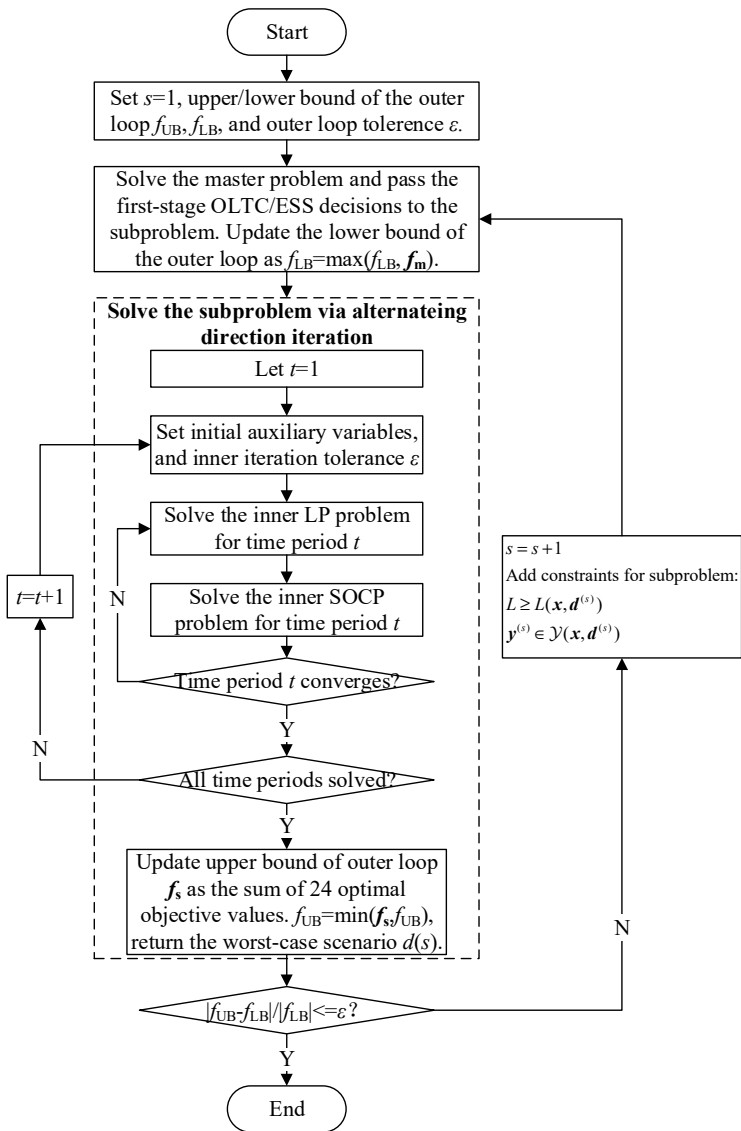

**Figure 3.** Algorithm flowchart.

## 5. Results

### 5.1. The 33-Bus Distribution Network

In this section, the modified 33-bus distribution network is used to test the proposed method.

#### 5.1.1. Simulation Settings

Load, PV, ESS and FDS configurations are shown in Figure 4. Flexible interconnection is realized among buses 8, 22 and 33 through a three-terminal FDS, whose capacity and transmission efficiency are 1.5 MVA and 98%, respectively. OLTC is installed between buses 1 and 2 with 11 taps ($\pm 5 \times 1\%$). The maximum permissible number of daily operations is five. Five PVs are installed at buses 4, 7, 16, 21 and 24 with capacity 2 MW. ESS is installed at bus 6. The capacity, maximum charging/discharging power, maximum/minimum SOC and charging/discharging efficiency are 3 MWh, 1000 kW, 90%/20% and 95%, respectively.

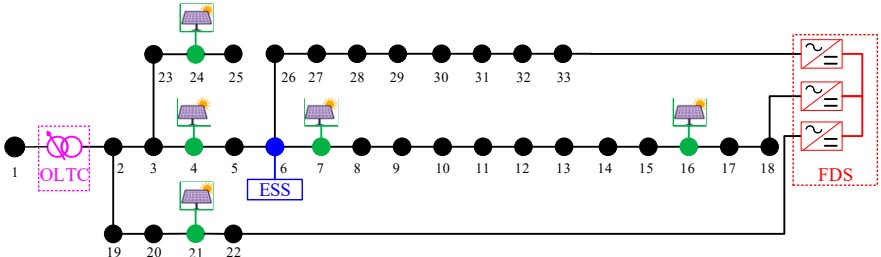

**Figure 4.** The 33-bus distribution network.

The proposed robust scheduling method based on the PWCH uncertainty set is compared with an existing robust scheduling method based on the polyhedral uncertainty set. The budget parameter $\Gamma$ is set to five, such that it is equivalent with the box uncertainty set, which encloses all historical data points. The PV outputs are set to fluctuate $\pm15\%$ from the forecasted values. The allowed range of voltage amplitudes is set to 0.95~1.05 p.u. Both methods are implemented using the AMPL modeling language [18] and the Gurobi 9.5.1 solver [19]. The test environment is a desktop computer with i7-9700 CPU, 2.40 GHz and 16 GB RAM.

5.1.2. Worst-Case Scenario Analysis

Figure 5 shows the historical data points of three PVs and the convex hull that enclose them. As seen from the figure, the convex hull uncertainty set is smaller in size compared to the minimal box uncertainty set that encloses this convex hull and has the potential to reduce decision conservativeness by cutting regions in the box with low probability of occurrence.

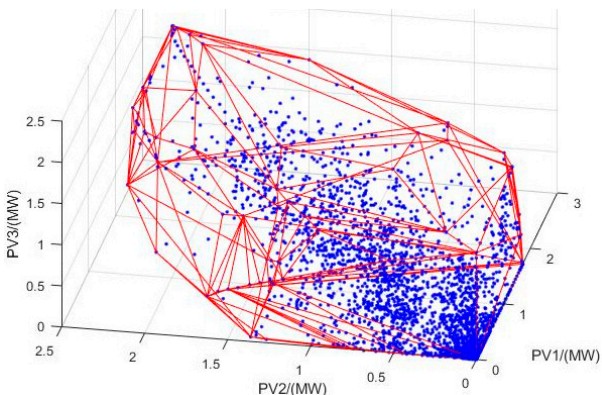

**Figure 5.** Schematic diagram of the 3D convex hull.

The worst-case scenarios selected by the CCG subproblem during the last iteration in the 8~16th time periods using the box uncertainty set and the PWCH uncertainty set are shown in Tables 3 and 4. It can be seen that the worst scenarios obtained by the two methods are different. Each worst scenario obtained by the box uncertainty set is a vertex of the box set of the corresponding time period, which corresponds to the upper limits of all PVs' outputs in that time period and cannot reflect the geographical correlation among the PVs' outputs. A 2-D illustration is shown in Figure 6, which intuitively describes the different worst scenarios obtained using two uncertainty sets. On the other side, each worst scenario selected by the PWCH uncertainty set is much worse, which, in fact, corresponds to a vertex of the intersection of all PWCHs in that time period. In other words, the worst scenario obtained from the box uncertainty set is worse and has a much lower probability of occurrence. If it is added to the CCG master problem, a more costly and conservative scheduling decision will be obtained, while if the worst scenario obtained by the PWCH uncertainty set is added to the CCG master problem, the scheduling decision will be less costly and conservative.

**Table 3.** Worst scenarios using the PWCH uncertainty set.

| Time Period | PV1 | PV2 | PV3 | PV4 | PV5 |
|---|---|---|---|---|---|
| 8 | 0.684169 | 0.681293 | 0.679652 | 0.675548 | 0.671436 |
| 9 | 1.227029 | 1.227765 | 1.213053 | 1.221142 | 1.196871 |
| 10 | 1.552959 | 1.558479 | 1.534559 | 1.548357 | 1.50604 |
| 11 | 1.746943 | 1.741806 | 1.731539 | 1.742832 | 1.698664 |
| 12 | 1.950833 | 1.902491 | 1.940068 | 1.947342 | 1.883357 |
| 13 | 2.171167 | 2.089821 | 2.13277 | 2.068701 | 2.037691 |
| 14 | 1.655152 | 1.650435 | 1.668362 | 1.609861 | 1.563627 |
| 15 | 1.37165 | 1.347866 | 1.36587 | 1.360097 | 1.320059 |
| 16 | 0.868731 | 0.86399 | 0.869861 | 0.867248 | 0.851022 |

**Table 4.** Worst scenarios using the Box uncertainty set.

| Time Period | PV1 | PV2 | PV3 | PV4 | PV5 |
|---|---|---|---|---|---|
| 8 | 0.708400 | 0.708400 | 0.708400 | 0.708400 | 0.708400 |
| 9 | 1.268833 | 1.268833 | 1.268833 | 1.268833 | 1.268833 |
| 10 | 1.587000 | 1.587000 | 1.587000 | 1.587000 | 1.587000 |
| 11 | 1.771767 | 1.771767 | 1.771767 | 1.771767 | 1.771767 |
| 12 | 2.008667 | 2.008667 | 2.008667 | 2.008667 | 2.008667 |
| 13 | 2.246333 | 2.246333 | 2.246333 | 2.246333 | 2.246333 |
| 14 | 1.627633 | 1.627633 | 1.627633 | 1.627633 | 1.627633 |
| 15 | 1.344733 | 1.344733 | 1.344733 | 1.344733 | 1.344733 |
| 16 | 0.838733 | 0.838733 | 0.838733 | 0.838733 | 0.838733 |

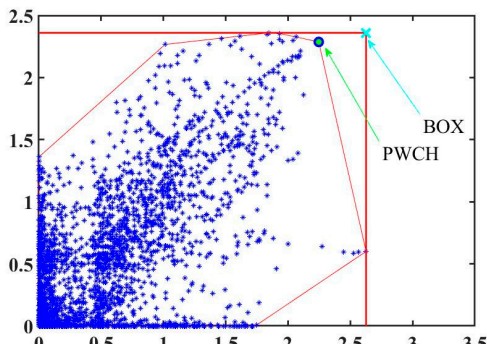

**Figure 6.** Comparison of the worst scenarios using two uncertainty sets.

### 5.1.3. OLTC Scheduling Strategy Analysis

Figures 7 and 8 show the OLTC action strategy and the ESS SOC for each time period obtained by the two scheduling methods, respectively. It can be seen that:

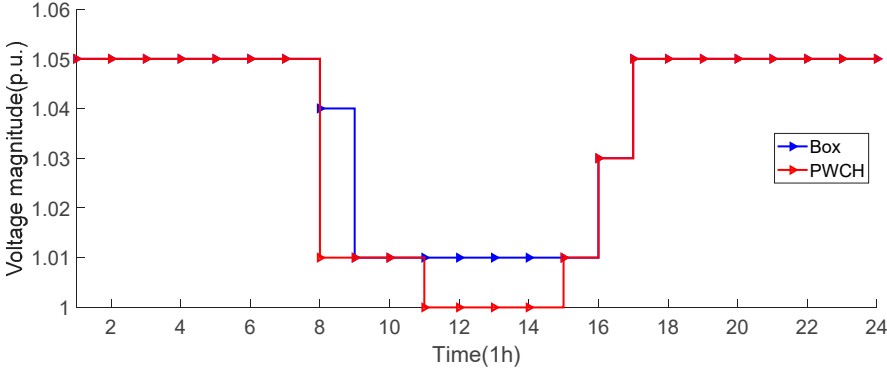

**Figure 7.** OLTC strategy comparison using two scheduling methods.

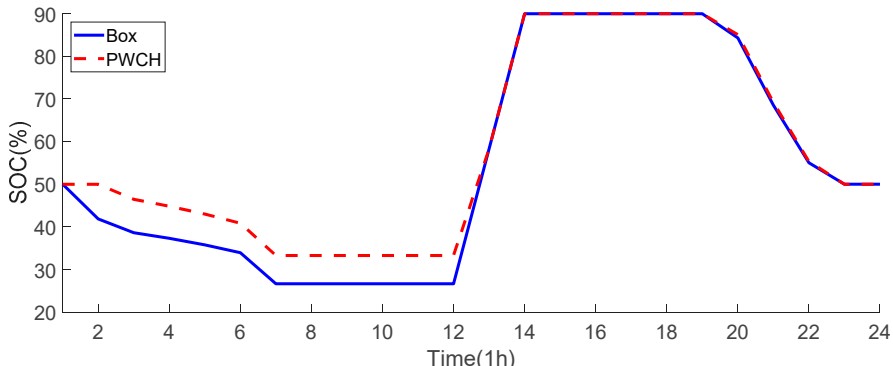

**Figure 8.** SOC comparison of the ESS using two scheduling methods.

(1) During time periods 0~6, the ESS discharges and the OLTC tap position is adjusted to the highest level so that the voltage at each bus is higher than the allowed lower limit;

(2) During time periods 8~11, the OLTC turns down the tap position. This is because the PV outputs increase during these time periods and there is a risk that the voltage of the PV buses violate the upper limit. Therefore, the OLTC cooperates with the FDS to absorb the PV power. As the efficiency of the ESS is lower than the FDS, the ESS SOC remains unchanged, and the excessive PV power is only consumed by cooperated control of the OLTC and the FDS;

(3) During time periods 12~16, the PV output reaches the maximum, and the cooperation between OLTC and FDS cannot achieve full PV consumption, so the ESS joins to cooperate with the FDS to absorb the excessive real power and avoid the risk of voltage violation;

(4) During time periods 17~24, the PV output gradually decreases to zero, while the load increases during these periods, so the OLTC turns down the tap position and cooperates with the FDS to consume the PV output;

(5) During time periods 17~24, the PV output gradually decreases to zero, and the load continues to increase during these periods, so the OLTC tap position is adjusted back to a high level, while the ESS discharges to supply the load consumption.

5.1.4. Analysis of Bus Voltage Levels

The bus voltage levels at each time period under the two scheduling methods are shown in Figure 9. It can be seen that during all time periods, the voltage level of each bus obtained by the two scheduling methods are within the safe range, thanks to the coordination of the OLTC, ESS and FDS.

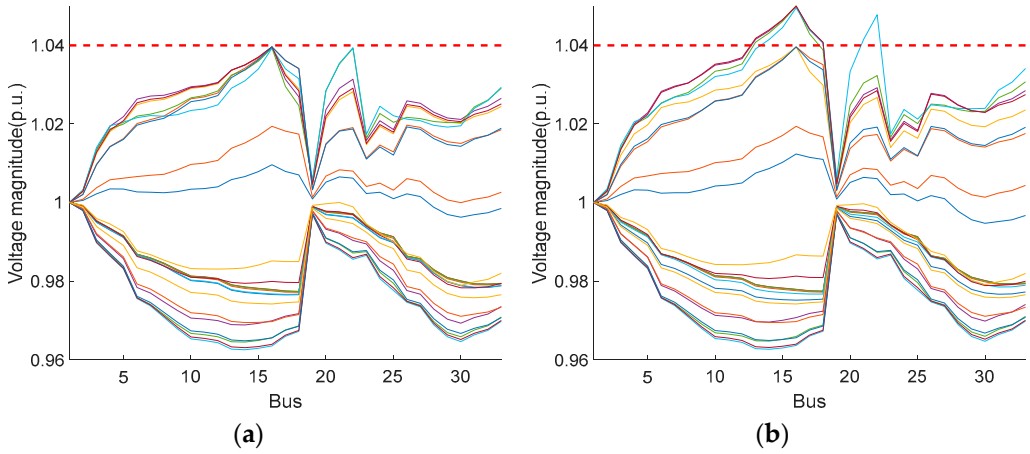

**Figure 9.** Bus voltage levels using two scheduling methods. (**a**) Box uncertainty set; (**b**) PWCH uncertainty set.

### 5.1.5. FDS Scheduling Strategy Analysis

The scheduling strategies of the FDS using two scheduling methods for each time period are shown in Figure 10. Combining the ESS and OLTC scheduling strategies, it can be seen that:

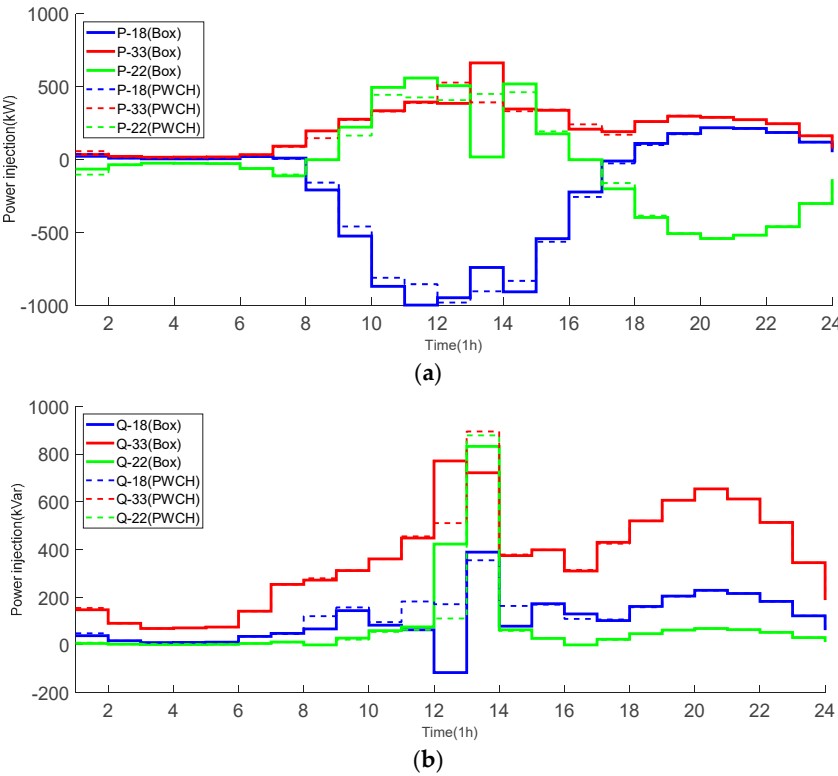

**Figure 10.** FDS power strategies using two scheduling methods. (**a**) real power; (**b**) reactive power.

(1) The FDS control strategies obtained by the two scheduling methods have basically the same trend during the PV unproductive periods, while there is a big difference between the two FDS scheduling strategies during time periods 10~13;

(2) During time periods 8~17, the FDS mainly extracts real power from lightly loaded side at bus 18 and injects them into the heavily loaded side at bus 22, 33 so that the loading levels of both sides can be balanced, and the voltage violations can be removed;

(3) During most time periods, both methods let the FDS inject reactive power to all sides to support the voltage levels;

(4) Compared with the PWCH-based method, the box-based method transmits more real/reactive power and makes a more conservative decision.

### 5.1.6. Daily Network Loss Comparison

The overall results of the two scheduling methods are shown in Table 5. It can be seen that the scheduling strategy based on the box uncertainty set is more conservative than the optimal scheduling strategy based on the PWCH uncertainty set in terms of network loss, FDS loss and ESS loss. The reason is that the worst scenarios selected by the box uncertainty set are much worse that scarcely happen, so a more conservative scheduling strategy is made to handle this scenario. At the same time, the more conservative scheduling strategy also loses some economy and flexibility. In addition, the algorithm time of both methods are comparable in terms of computational efficiency.

**Table 5.** Results comparison of two scheduling methods.

|  | PWCH | Box |
|---|---|---|
| Objective function (MW) | 2.783 | 3.695 |
| CPU time (s) | 483 | 511 |
| Iterations number | 2 | 2 |
| ESS loss (MW) | 0.175 | 0.195 |
| FDS loss (MW) | 0.492 | 0.513 |
| Network loss (MW) | 2.115 | 2.987 |

### 5.2. A Realistic 104-Bus Distribution Network

In this section, a larger 104-bus distribution network is used to test the proposed method.

#### 5.2.1. Simulation Settings

The system comes from a realistic 10 kV distribution network from the China Southern Grid. Load, PV, ESS and FDS configurations are shown in Figure 11. The maximum and minimum real and reactive load are 22.26 MW and 10.48 MVar, respectively. Flexible interconnection is realized among buses 17, 46, 53 and 104 through a four-terminal FDS, whose capacity and transmission efficiency are 3 MVA and 98%, respectively. Five PVs are installed at buses 9, 32, 58, 91 and 92 with capacity 3 MW. Four ESSs are installed at buses 7, 56, 76 and 90. The capacity, maximum charging/discharging power, maximum/minimum SOC and charging/discharging efficiency are 1.5 MWh, 300 kW, 90%/20% and 97%, respectively. Similarly with the previous section, robust scheduling methods based on the PWCH uncertainty set and the box uncertainty set are compared under the same environment with that used by the 33-bus distribution network.

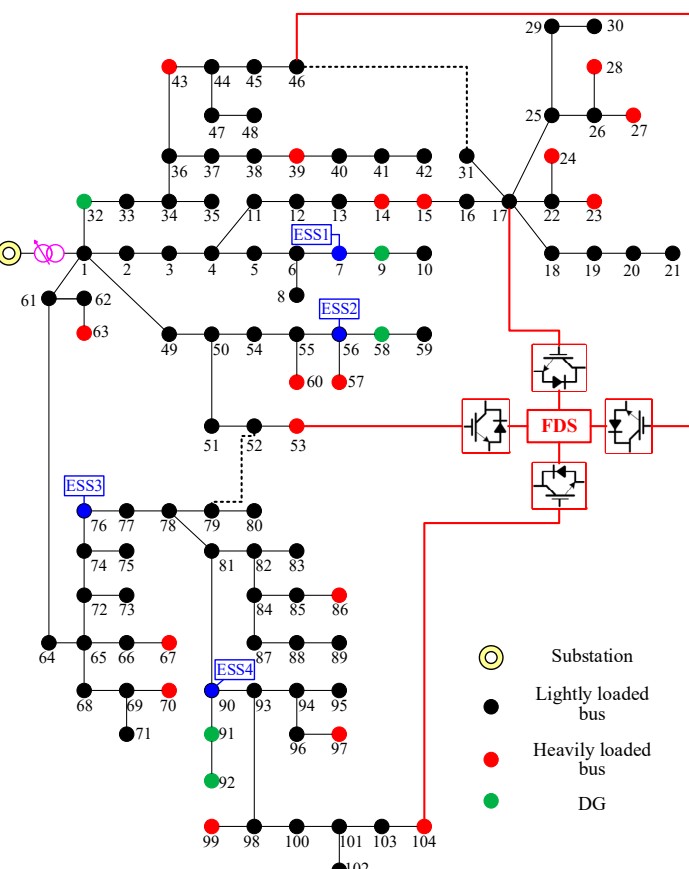

**Figure 11.** The realistic 104-bus distribution network.

### 5.2.2. OLTC/ESS Scheduling Strategy Analysis

Figures 12 and 13 show the OLTC action strategy and the ESS SOC of the 104-bus system using two scheduling methods, respectively. It can be seen that:

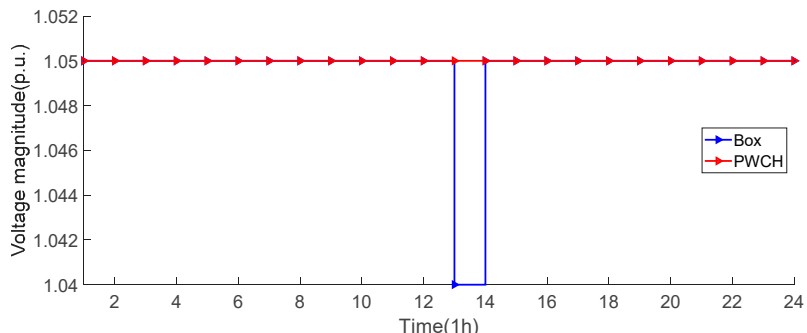

**Figure 12.** OLTC strategy comparison using two scheduling methods (the 104-bus system).

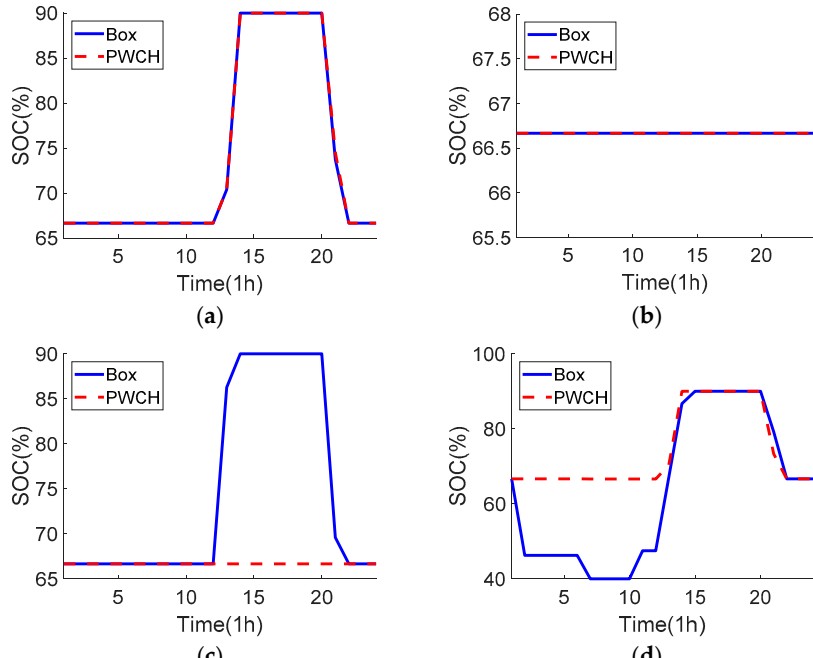

**Figure 13.** SOC comparison of the ESSs using two scheduling methods (the 104-bus system). (**a**) ESS 1 at bus 7; (**b**) ESS 2 at bus 56; (**c**) ESS 3 at bus 76; (**d**) ESS 4 at bus 90.

(1) For ESS 1 at bus 7, both scheduling methods give the same charging/discharging strategies, which coincides with the loading patterns and PV output patterns such that peak shifting (the 13th time period when the PV output is the largest) and valley filling (the 20th time period when the load level is the heaviest) can be achieved;

(2) For ESS 2 at bus 56, both scheduling methods prefer to not charge/discharge this ESS due to operation costs when no voltage violation exits;

(3) For ESS 4 at bus 90, as this ESS is very close to PVs, the box-based strategy decides to release ESS 4's energy in advance during time periods 1~10 to prepare for absorbing excessive PV output in time period 13. However, discharging ESS 4 only is not enough to fully prevent any voltage violation risk, thus the OLTC turns down the tap position during time period 13 and cooperates with the upstream ESS 3 at bus 76 to help consume the excessive PV output. On the other side, the PWCH-based strategy neither discharge ESS 4 in advance nor operate the OLTC. The upstream ESS 3 at bus 76 also does not charge/discharge for all time periods. This is because the worst scenario selected from

the PWCH uncertainty set is not such harsh as that selected from the box uncertainty set. Therefore, the control strategy is less conservative.

In summary, the scheduling strategy based on the PWCH uncertainty set is clearly more flexible and leads to lower ESS losses.

### 5.2.3. Analysis of Bus Voltage Levels

The bus voltage levels at each time period under the two scheduling methods are shown in Figure 14. It can be seen that the voltages of all buses fall in acceptable ranges using both scheduling methods, although both strategies can find the optimal operation strategy to satisfy the voltage level constraints. However, it is obvious that the voltage profile based on the PWCH uncertainty set is smoother, i.e., it is closer to the rated voltage level of the system.

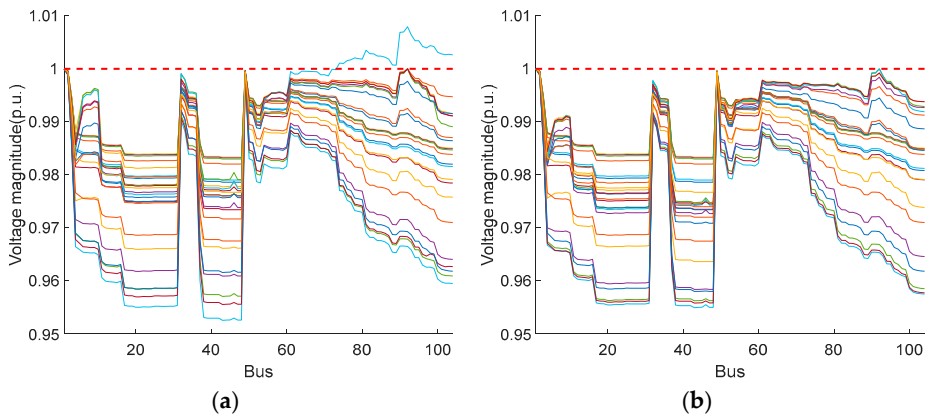

(a)        (b)

**Figure 14.** Bus voltage levels using two scheduling methods (the 104-bus system). (**a**) Box uncertainty set; (**b**) PWCH uncertainty set.

### 5.2.4. FDS Scheduling Strategy Analysis

The FDS scheduling strategies of the 104-bus system using two scheduling methods are shown in Figure 15. It can be seen that:

(1) Both scheduling methods give the same results during the PV unproductive time periods 1~8, i.e., the FDS does not need to transmit any real/reactive power, while there is a big difference between the two FDS scheduling strategies during PV productive time periods 10~13 and heavily loaded time periods 17~23. The real power scheduling shows an obvious bimodal distribution with the fluctuation of load and PV output, and coincides with the bimodal peak of load fluctuation;

(2) During time periods 10~13, the FDS mainly extracts real power from lightly loaded side at bus 104 and injects them into the heavily loaded side at bus 46 so that the loading levels of both sides can be balanced, and the voltage violations can be removed. The FDS action strategy based on the PWCH uncertainty set only starts to act at time period 10, while the FDS action strategy based on the box uncertainty set already starts to act at time period 8 (just after the appearance of light). The FDS control strategy based on the box uncertainty set reaches the transmission limit at the 13th time period, while the FDS control strategy based on the PWCH uncertainty set reaches the maximum transmission power but does not reach the transmission limit at this time. Although the FDS control strategy based on the box uncertainty set reduces the voltage peaks at the 13~14th time period by increasing the real power transfer, it still cannot make the voltages at the 13~14th time period reach the ideal range due to the transmission capacity limit and the high PV output at noon, so the ESS charging power also reaches the maximum.

(3) During time periods 17~23, the FDS mainly extracts real power from lightly loaded side at bus 53 and injects them into the heavily loaded side at bus 17, 46 so that the loading levels of all sides can be balanced, and the voltage violations can be removed;

(4) During most time periods, both methods let the FDS inject reactive power to all sides to support the voltage levels; Unlike the real power strategy of the FDS, the PWCH uncertainty set-based FDS control strategy provides more reactive power compared to the box uncertainty set-based FDS control strategy when the PV output does not reach a higher maximum.

(5) Compared with the PWCH-based method, the box-based method transmits much more real/reactive power and makes more conservative decisions.

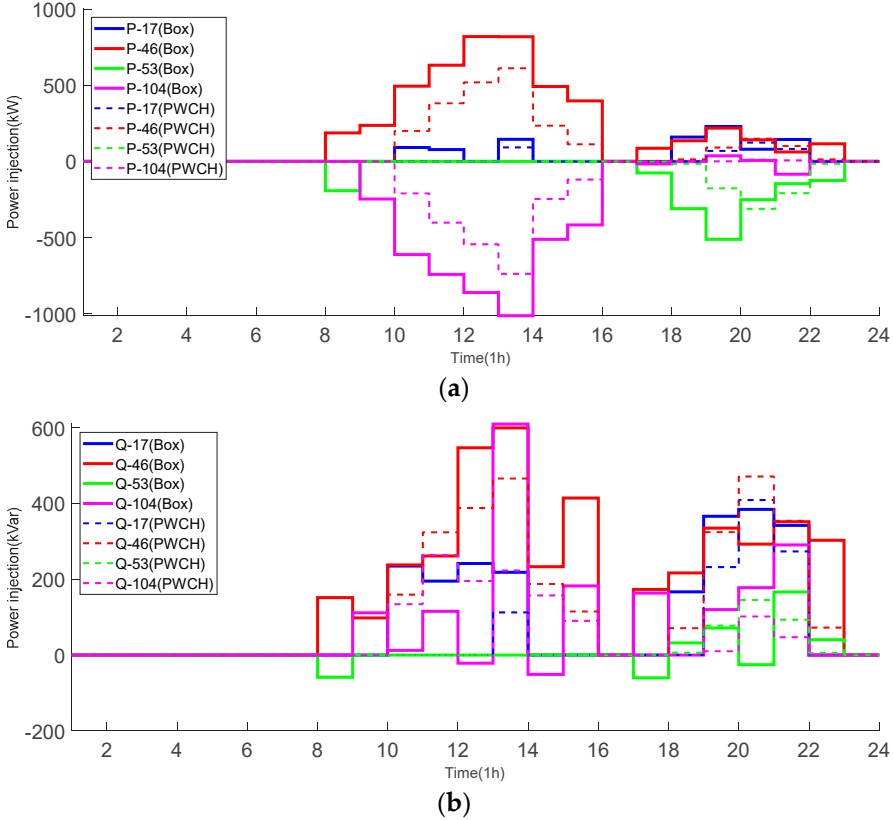

**Figure 15.** FDS power strategies using two scheduling methods (the 104-bus system). (**a**) real power; (**b**) reactive power.

### 5.2.5. Daily Network Loss Comparison

Finally, the overall results of the 104-bus system using both scheduling methods are shown in Table 6. Similarly, it can be seen that the scheduling strategy based on the box uncertainty set is more conservative than the scheduling strategy based on the PWCH uncertainty set in terms of network loss, FDS loss and ESS loss. Specifically, the decision scheme with PWCH uncertainty set reduces the network loss by 6.53% compared with the decision scheme with box uncertainty set, which verifies again the effectiveness of the PWCH uncertainty set in reducing decision conservativeness and improving economy and flexibility over the box uncertainty set.

**Table 6.** Results comparison of two scheduling methods (the 104-bus system).

|  | PWCH | Box |
| --- | --- | --- |
| Objective function (MW) | 4.0734 | 4.3914 |
| Iterations number | 2 | 2 |
| ESS loss (MW) | 0.0281 | 0.0823 |
| FDS loss (MW) | 0.1713 | 0.2976 |
| Network loss (MW) | 3.8740 | 4.0115 |

## 6. Conclusions

In this paper, a two-stage RO model is established for flexible distribution networks based on a novel PW-CH uncertainty set, considering the coordination of energy storage systems, OLTCs and FDSs. The temporal correlated OLTCs and ESSs are globally scheduled in the first stage using day-ahead forecasted DG outputs, while FDSs are scheduled in the second stage in real time in each time period based on the first-stage decisions and accurate short-term forecasted DG outputs. An improved column-and-constraint generation algorithm is used to solve the RO model in an efficient manner. Specifically, the geographical correlation of the outputs of multiple DGs is modeled based on the PWCH, such that the decision conservativeness can be reduced.

Test results show that the worst scenarios selected from the box uncertainty set are much worse but scarcely happen. Therefore, a more conservative scheduling strategy should be made to handle this scenario, resulting in higher operation costs and low flexibility. On the other side, by cutting regions in the box with low probability of occurrence, the proposed PWCH-based method can obtain much less conservative scheduling schemes with much lower operation costs.

In the future, more advanced optimization algorithms, including hybrid heuristics and metaheuristics, adaptive algorithms, self-adaptive algorithms and island algorithms [20–25], can be used for scheduling of flexible resources in distribution systems with better performance.

**Author Contributions:** Conceptualization, D.L. and H.Y.; methodology, D.L. and Z.W.; software, D.L. and Z.W.; validation, Z.W. and S.Y.; formal analysis, D.L.; investigation, D.L. and H.Y.; resources, D.L. and H.Y.; data curation, H.Y. and S.Y.; writing—original draft preparation, D.L. and Z.W.; writing—review and editing, D.L. and Z.W.; visualization, D.L. and Z.W.; supervision, D.L.; project administration, H.Y. and S.Y.; funding acquisition, H.Y. and S.Y. All authors have read and agreed to the published version of the manuscript.

**Funding:** This work was funded by the State Grid Hebei Electric Power Company grant number kj 2021-007.

**Institutional Review Board Statement:** Not applicable.

**Informed Consent Statement:** Not applicable.

**Data Availability Statement:** The original contributions presented in the study are included in the article, further inquiries can be directed to the corresponding author.

**Conflicts of Interest:** Authors H.Y. and S.Y. are employed by State Grid Hebei Electric Power Company Hengshui Power Supply Company. Author H.Y. is employed by Hengshui Electric Power Design Co., Ltd. The remaining authors declare that the research was conducted in the absence of any commercial or financial relationships that could be construed as a potential conflict of interest.

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
