# Peer review of "Two-Stage Robust Optimal Scheduling of Flexible Distribution Networks Based on Pairwise Convex Hull"

_sustainability, doi:10.3390/su15076093_

Round 1

Reviewer 1 Report

I recommend minor grammar improvements throughout the article. Figures 8 A and B are very small. You can't read anything from them.

I recommend the authors revise the conclusion of the article. I would expand and clarify the conclusion.

Questions for the authors:

Have you applied the algorithm to a different network than the one presented? What were the results?

Have you verified this optimization in the real world?

How scalable is the optimization?

Is the PV data from real-world traffic, or are you confused by the simulated data? Where is the network/model you are doing the optimization on? What time of year is the simulation done?

Are you comparing two methods to each other? What other methods are current, and what are the results?

I recommend the article with the minor modifications I have mentioned.

Thank you

Reviewer 2 Report

First of all, congratulate the authors for the research paper submitted for publication in this journal.

The abstract does not clearly indicate the contributions and novelties of the authors to the study problem analyzed in the document. Some of these details appear throughout the document. In this way, the advances, results, conclusions, and importance of the research carried out are not well appreciated. It is advisable to rewrite the summary indicating the main contributions, as well as the conclusions obtained.

The introduction incorporates a brief state of the art. It is recommended that this section be much more detailed and include, for example, the main research studies and other issues associated with the study addressed. It may be interesting to make comparisons with other research papers already published, showing other techniques related to this topic. In this case, the main contributions and novelties of the authors to the problem analyzed have not been very clear, nor have the advantages and disadvantages compared to other investigations carried out. As a suggestion, perhaps the elaboration of a comparative table that compiles the contributions to the study of the main bibliographical references, can give clarity to this state-of-the-art. The number of bibliographical references cited in the document is adequate for the study carried out.

On the other hand, the document is technically sound, since it contains a brief analysis of the state of the art (as has been commented, it is necessary to rewrite this section) related to the study carried out. It also incorporates different concepts, details, and statistical information related to the development of the model. It also includes some estimated results that support the analysis shown.

The concepts have been presented comprehensively. The different figures, tables, diagrams, and attached schemes facilitate the understanding of the contents presented by the authors in the document. The only drawback is that some figures are so small that it is impossible to appreciate the details inside. Likewise, the simulated results obtained support the comments made by the authors. All this makes it easier for the reader to follow the document.

Most of the text incorporated in the conclusions section is more geared towards a discussion section than the conclusions themselves. It would be advisable to rewrite this section and include a section for the discussion of the results. The conclusions section must include the main ideas, contributions, and results obtained by the authors of the study.

Finally, it is advisable to calmly read the text in English or have a native English speaker review it, as words appear in the document that is not commonly used.

Reviewer 3 Report

I would like to thank the authors for their work. However, there are some comments need to be addressed.

1- In my opinion, the introduction has some sort of ambiguity. Each paragraph must be linked with the previous in a correlated way.

2- The main contribution of the paper is not clear. this must be clearly mentioned in the abstract and in the conclusion.

3- Some figures are impossible to see.

4- In my opinion, the method need to be tested in case of FRT, especially for inverter based DG. Could the authors provide at lease one case for this scenario. 

Round 2

Reviewer 2 Report

First of all, congratulate the authors for the research work carried out and presented in this journal for publication. The authors have incorporated into the document the different comments, details, and observations made by the reviewers. In this way, the summary has been rewritten indicating the main novelties and contributions of the authors. Likewise, the state-of-the-art has been improved by incorporating comparative tables and new references. Finally, observations have been incorporated in some sections, according to the comments and details of the reviewers, and the conclusions have been rewritten.

Reviewer 3 Report

Thanks to the authors for their answers.
